# Amyloid and Tau Protein Concentrations in Children with Meningitis and Encephalitis

**DOI:** 10.3390/v14040725

**Published:** 2022-03-30

**Authors:** Artur Sulik, Kacper Toczylowski, Agnieszka Kulczynska-Przybik, Barbara Mroczko

**Affiliations:** 1Department of Pediatric Infectious Diseases, Medical University of Bialystok, Waszyngtona 17, 15-274 Bialystok, Poland; kacper.toczylowski@umb.edu.pl; 2Department of Neurodegeneration Diagnostics, Medical University of Bialystok, Waszyngtona 15A, 15-269 Bialystok, Poland; zdchn@umb.edu.pl (A.K.-P.); mroczko@umb.edu.pl (B.M.)

**Keywords:** amyloid-beta, tau protein, Alzheimer’s disease biomarkers, meningitis, children

## Abstract

Alzheimer’s disease (AD) has emerged as a growing threat to human health. It is a multifactorial disorder, in which abnormal amyloid beta metabolism and neuroinflammation have been demonstrated to play a key role. Intrathecal inflammation can be triggered by infections and precede brain damage for years. We analyzed the influence of infections of the central nervous system on biomarkers that are crucially involved in AD pathology. Analyses of the cerebrospinal fluid (CSF) levels of Aβ_1–42_, Aβ_1–40_, Tau, and pTau proteins were performed in 53 children with neuroinfections of viral (*n* = 26) and bacterial origin (*n* = 19), and in controls (*n* = 8). We found no changes in CSF amyloid Aβ_1–42_ concentrations, regardless of etiology. We showed an increase in tau and phosphorylated tau concentrations in purulent CNS infections of the brain, compared to other etiologies. Moreover, the total concentrations of tau in the CSF correlated with the CSF absolute number of neutrophils. These findings and the Aβ 42/40 concentration quotient discrepancies in CFS between meningitis and encephalitis suggest that infections may affect the metabolism of AD biomarkers.

## 1. Introduction

Alzheimer’s disease (AD) is the most common neurodegenerative disease, and causes memory loss, cognitive decline, and crucially interferes with patients’ everyday lives. The hallmarks of AD include extracellular plaques, due to the deposition of amyloid-beta (Aβ) peptides, and intracellular neurofibrillary tangles, resulting from tau phosphorylation. Understanding Aβ and tau accumulation and clearance is crucial, in order to recognize AD pathways and develop future therapies. Cerebral amyloid angiopathy is another condition linked to Aβ deposition in the brain. Both diseases are driven by impaired Aβ clearance and share pathogenic implications [1]. Aβ is normally produced by hydrolyzing amyloid precursor proteins in brain parenchyma. The following two subtypes of Aβ can be identified in the human body: Aβ_1–40_ and Aβ_1–42_. Extracellular Aβ clearance is accomplished by glial phagocytosis, involving microglia, astrocytes, and protease activity from neurons and astrocytes [2].

Numerous factors have been designated to promoting cerebral amyloid accumulation. Among them, aging and several genetic risk factors are well recognized [3]. Currently, neuroinfection has emerged as a relevant element leading to protein accumulation and aggregation in the brain [4]. A number of viruses are known for their neurotropism. Many enteroviruses and most members of the *Herpesviridae* family, such as herpes simplex virus-1 and -2 (HSV-1, -2), varicella zoster virus (VZV), cytomegalovirus (CMV), and Epstein–Barr virus (EBV), are capable of reaching the central nervous system (CNS). Herpesviruses were first proposed to contribute to AD pathology [5]. Herpes simplex virus-1 (HSV-1) remains latent in neurons [6], staying dormant for years, but eventually it can reactivate and replicate in a lytic cycle that produces infectious virions [7]. Controlling HSV-1 throughout life involves the immune system; thus, episodes of immune depression and concomitant infections may trigger virus reactivation and replication. Infection with HSV-1 leads to the accumulation of key AD proteins in the human tissue model, and this process can be reduced by anti-herpes antiviral agents [8]. An increased abundance of human herpesvirus 6A (HHV-6A) and human herpesvirus 7 (HHV-7) was detected in the brain tissue of subjects with AD, compared to controls, strengthening the role of microbes in the onset and progression of AD [9]. Human immunodeficiency virus (HIV) is another virus with a possible link to increased beta-amyloid synthesis, leading to neurodegeneration and Alzheimer’s disease [10].

The role of bacteria in AD pathways is uncertain. The immunoreactivity of *Chlamydia pneumoniae* antigens was observed both intra- and extracellularly in AD brains [11]. Experimentally, amyloid deposits similar to AD plaques were formed in the brains of mice following infection with *Chlamydia pneumoniae* [12]. Morphological changes, corresponding to the amyloid deposits of AD, were observed following exposure to *Borrelia* spirochetes [13].

The pathways leading to Alzheimer’s disease begin many decades before the diagnosis of AD dementia [14]. It was postulated that antibody-mediated therapies for Alzheimer’s disease, as well as any future therapeutic vaccines, should be launched early in the prodromal or symptom-free stage of the disease, in order to be effective [15]. Performing studies in the early stage of life may contribute to a better understanding of the pathophysiological process of AD. In our study, we aimed to investigate whether CNS infections could contribute to amyloid and tau pathology in children. We measured CSF concentrations of Aβ and tau in various viral neuroinfections, and compared them with bacterial infections and controls.

## 2. Materials and Methods

A total of 53 children were included in the study. All the patients in the study and in the control group were referred to the hospital with acute febrile illness and did not have any underlying chronic disorders. In twenty-six children, a viral cause of the infection was detected. This included 12 children with an infection caused by enteroviruses, 5 children with an infection caused by the varicella zoster virus (VZV), and 9 children with an infection caused by the tick-borne encephalitis virus (TBEV). The control group comprised 8 febrile children suspected of having meningitis, in whom the diagnosis was ruled out and who were eventually diagnosed with a non-specific viral infection. To broaden the analysis of the impact of CNS infection on biomarkers of AD, we also included 19 children with two types of bacterial infections of the CNS—purulent and aseptic. In the purulent group, there were eight children with infections caused by *Neisseria meningitidis* (*n* = 2), *Streptococcus pneumoniae* (*n* = 2), *Streptococcus agalactiae* (*n* = 2), *Haemophilus influenzae* (*n* = 1), and *Escherichia coli* (*n* = 1). The bacterial aseptic group comprised 11 children diagnosed with Lyme neuroborreliosis caused by *Borrelia burgdorferi*.

In the study, meningitis was defined as symptoms consistent with meningitis and a leucocyte count in the cerebrospinal fluid above 5 cells per µL. Encephalitis was defined, according to the International Encephalitis Consortium, as an altered mental status (defined as a decreased or altered level of consciousness, including change in personality and lethargy) for over 24 h, with no alternative cause identified and two of the following: seizures, focal neurologic findings, abnormalities in electroencephalography or magnetic resonance imaging suggestive of encephalitis, cerebrospinal fluid (CSF) pleocytosis, and fever. Enteroviruses and VZV were detected in the CSF by polymerase chain reactions. The diagnosis of TBE and LNB was confirmed with serological assays. Purulent infections were diagnosed based on positive CSF culture.

The biochemical measurements of neurochemical dementia diagnostic (NDD) biomarkers, such as Aβ_1–42_, Aβ_1–40_, tau, and pTau181, in the CSF were performed in the Department of Neurodegeneration Diagnostics, Medical University of Bialystok, Poland. The concentrations of AD biomarkers were assessed with a commercially available quantitative enzyme-linked immunosorbent assay, using IBL kits (IBL International, Hamburg, Germany) for Aβ_1–42_ and Aβ_1–40_, and Fujirebio kits (Fujirebio Europe, Gent, Belgium) for tau and pTau181 proteins. The standards and samples were run in duplicates, with a coefficient of variance (CV) < 20%. The assays were performed following the manufacturer’s instructions, and CSF samples were diluted 1:20 for Aβ_1–42_ and Aβ_1–40_, and were undiluted for tau and ptau.

The summary statistics for continuous variables are presented as a median with interquartile range (IQR); categorical variables are presented as frequencies. The differences between groups were analyzed with the one-way analysis of variance (ANOVA), followed by Newman–Keuls post hoc test. Non-parametric tests were used for other analyses. The results were considered statistically significant when the *p*-value was less than 0.05. The statistical analysis was performed with the use of TIBCO Software Inc. (2017) Statistica, version 13 (Palo Alto, CA, USA).

## 3. Results

The characteristics of the study groups are presented in Table 1.

We found that infections of the CNS had a limited effect on the total amyloid concentrations and on the ratio of Aβ_1–42_ to Aβ_1–40_ in the CSF, regardless of etiology (Table 2). The total and phosphorylated tau concentrations were markedly increased in purulent bacterial infections of the CNS, compared to other etiologies (Figure 1).

The comparison of purulent and aseptic (viral and bacterial) CNS infections yielded similar results (Figure 2). In purulent infections, the concentrations of total and phosphorylated tau were markedly higher, compared to the controls and the aseptic group. The concentrations of Aβ_1–40_ were reduced, but Aβ_1–42_ and the amyloid ratio were not affected. The ratios of total and phosphorylated tau to amyloid β isoforms were increased in purulent CNS infections (Appendix A). The Spearman’s R analysis found no significant correlation between total and phosphorylated tau and all of the CSF parameters. In a stepwise regression analysis, we found that the concentrations of total tau in the CSF were correlated with the absolute number of neutrophils in the CSF (R^2^ = 0.11, *p* = 0.02).

In the comparison between viral and bacterial (including both purulent infections and Lyme neuroborreliosis) CNS infections, no significant differences were found in the levels of the biomarkers. 

To further analyze the impact of CNS infections on biomarkers of AD, we compared children with viral meningitis and viral encephalitis. Children diagnosed with viral meningitis, compared to children with viral encephalitis, had a lower Aβ_42/40_ ratio (Figure 3). The concentrations of total tau, phosphorylated tau, and tau/amyloid β ratios did not differ in this comparison (Appendix A).

In the search for possible causes of the reduced Aβ_42/40_ ratio in children with meningitis, we performed a stepwise regression analysis. In the model, we included the CSF parameters (CSF protein, CSF cell count, CSF lymphocyte count, CSF neutrophil count, and CSF monocyte count). No significant correlations were found between amyloid isoforms, amyloid ratio, and the CSF parameters. In addition, the Spearman’s R analysis found no significant correlation between the Aβ_42/40_ ratio, amyloid isoforms, and all of the CSF parameters.

## 4. Discussion

In our study, we analyzed the CSF concentration of Alzheimer’s disease biomarkers in CSF in children with central nervous system infections of different etiologies. The distribution of etiologies in the study group reflects the epidemiology of CNS infections in our region. Northeastern Poland is an endemic area for tick-borne diseases, including tick-borne encephalitis [16,17] and Lyme disease [18]. This explains the considerably high number of cases of tick-borne diseases in the study group. Moreover, pneumococcal and meningococcal vaccines were not offered in the routine immunization schedule throughout the study period, resulting in the occurrence of invasive diseases caused by encapsulated bacteria in our population and the study group.

A reduced concentration of Aβ_1–42_ in CSF is one of the best-established biomarkers of AD [19]. Key regulators of the Aβ levels in the brain are the transport across the blood-brain barrier, proteolytic degradations, oligomerization, and aggregation. In addition, other neurovascular cells, such as astrocytes, produce clearly different forms of Aβ [20]. It has been proposed that a reduction in soluble Aβ in the CSF is caused by Aβ deposition in amyloid plaques in AD patients [21]. However, low Aβ_1–42_ levels may also be observed in infections of the CNS, where the abundance of plaques is low, suggesting downregulation of amyloid expression and/or processing in these diseases. Studies have found decreased Aβ_1–42_ levels in bacterial meningitis, multiple sclerosis, and HIV with cerebral engagement [22,23,24]. Studies of viral meningitis have exhibited conflicting results [22,25,26]. Additionally, low levels of Aβ_1–42_ were associated with worse outcomes of CNS infections [26,27]. Thus, mechanisms other than plaque accumulation could contribute to low CSF Aβ_1–42_ levels in diseases of the CNS. Presently, the cause of altered levels of amyloid isoforms in the CSF is not clear. In this study, we found no changes in the CSF concentrations of Aβ_1–42_ and Aβ_1–40_, regardless of etiology. The lack of statistical importance could be, to some extent, a result of the rather limited number of children included in each of the study subgroups. This is a result of the heterogeneity of infections observed during the study period. There was a trend towards lower concentrations of Aβ_1–42_ and Aβ_1–40_ in all the study groups; however, the differences did not reach statistical significance in all, but one, of the comparisons. In a further analysis, we found that the concentrations of Aβ_1–40_ were significantly lower in children with purulent meningitis, compared to the control group. In the search for a cause of altered amyloid isoform levels, we found no associations between the amyloids and the CSF parameters.

Currently, the knowledge of inflammation-induced changes in Aβ peptide metabolism is mainly based on studies of acute infections. In a small case series study, a very low level of Aβ_1–42_ was reported in tuberculous meningitis [28]. The concentrations measured were lower compared with both the control group and the AD group of patients. It should be noted that the initial symptoms of tuberculous meningitis are nonspecific, making diagnosis challenging. The time between the first symptoms and hospital admission can be weeks, or even months [28]. The changes in AD markers observed in tuberculous meningitis in adult patients at the time of diagnosis may reflect inflammatory processes that last for a long time. In contrast, in our study, we focused on etiologies causing much more acute clinical symptoms, and our CSF samples were collected in the early acute stage of the disease. Similarly, in a study of patients with Lyme neuroborreliosis, who were examined three weeks after the onset of symptoms, the concentrations of all the measured Aβ peptide variants were reduced, compared to the controls [29]. Thus, the etiology, duration, or severity of neuroinflammation might affect its influence on Aβ metabolism.

Recently, a reduced Aβ_42/40_ ratio in CSF was found to be more accurate in the diagnosis of AD than Aβ_1–42_ alone [30,31,32]. In AD patients, this decrease is caused by the decrease in Aβ_1–42_ and an increase in Aβ_1–40_ CSF concentrations [33]. We report a reduced Aβ_42/40_ ratio in children with viral meningitis, compared to children with viral encephalitis. However, the underlying cause was different than in AD, as we did not find an increase in any of the amyloid isoform levels. When compared to the controls, the median Aβ_1–40_ concentrations were reduced more in encephalitis than in meningitis patients, although the differences did not reach statistical significance. This might explain the observed variability in the relative concentration of amyloid isoforms observed in our study. It is noteworthy that most, if not all, of the research on Alzheimer biomarkers in various CNS infections is conducted in adult patients. The results of these studies are not the best reference for our research in children. 

Tau is an important part in the stability and regulation of the axonal cytoskeleton. Increased tau levels in the CSF reflect neuronal cell death, with the release of tau-related proteins into the CSF. Hyperphosphorylation of tau causes the proteins to detach from the microtubules and destabilizes them [34]. Importantly, the increased CSF tau levels observed in acute CNS pathologies, such as head trauma and infections, are likely to be a result of direct and extensive neuronal damage. However, in AD, the processes leading to tau pathology are complex, and involve the toxicity of misfolded tau species and interactions with amyloid proteins [35]. We documented an increase in tau and phosphorylated tau concentrations in purulent CNS infections compared to other etiologies, which reflects marked neuronal damage in bacterial meningitis. The measurement of tau protein concentrations in CSF could aid in distinguishing serious CNS infections of potentially poor outcome from benign meningitis. The tau protein is also involved in blood–brain barrier dysfunction, which is described in numerous neurological conditions [36], and possibly plays a role in infectious pathologies of the CNS. Other studies showed abnormally high tau protein levels in herpesvirus CNS infections, which is also potentially devastating in outcome [27]. The tau protein was analyzed in a series of CSF samples from patients with tick-borne encephalitis (TBE). Interestingly, the tau protein concentration was found to predict a complicated course of tick-borne encephalitis in adults [37]. Severe complications of TBE in children occur much less often than in adults [38]. In this study, the tau concentrations were not increased in children with TBE. We found that the total concentrations of tau in the CSF were correlated with the absolute number of neutrophils in the CSF. Neutrophils are the most abundant leukocytes and the innate immune system’s first line of defense. The expression of specific neutrophil proteins may regulate neuroinflammation associated with AD [39]. For instance, neutrophil depletion decreased the levels of phosphorylated tau in the AD mice [40].

One of the limitations of this study is the relatively small sample size in each of the study groups. In addition, the concentration of AD biomarkers displayed high variation between individuals, which might have affected the statistical analysis. Nevertheless, to the best of our knowledge, this is the first study evaluating the biomarkers of AD in children with infections of the CNS, caused by different viral and bacterial pathogens. Our results confirm the notion that infections may, in fact, impact amyloid metabolism, but further investigation is needed, in order to gain a more complete understanding of this finding.

## Figures and Tables

**Figure 1 viruses-14-00725-f001:**
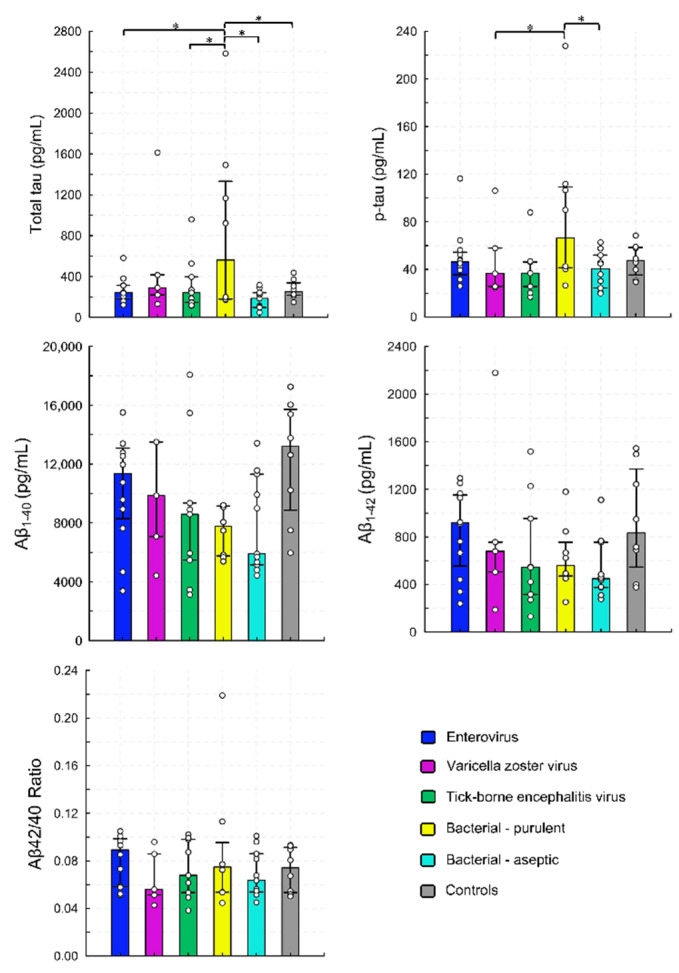
Cerebrospinal fluid concentrations of biomarkers of Alzheimer’s disease in children with infections of the central nervous system. Bars represent medians, whiskers represent interquartile range, and dots show the raw data. * indicates *p* < 0.05.

**Figure 2 viruses-14-00725-f002:**
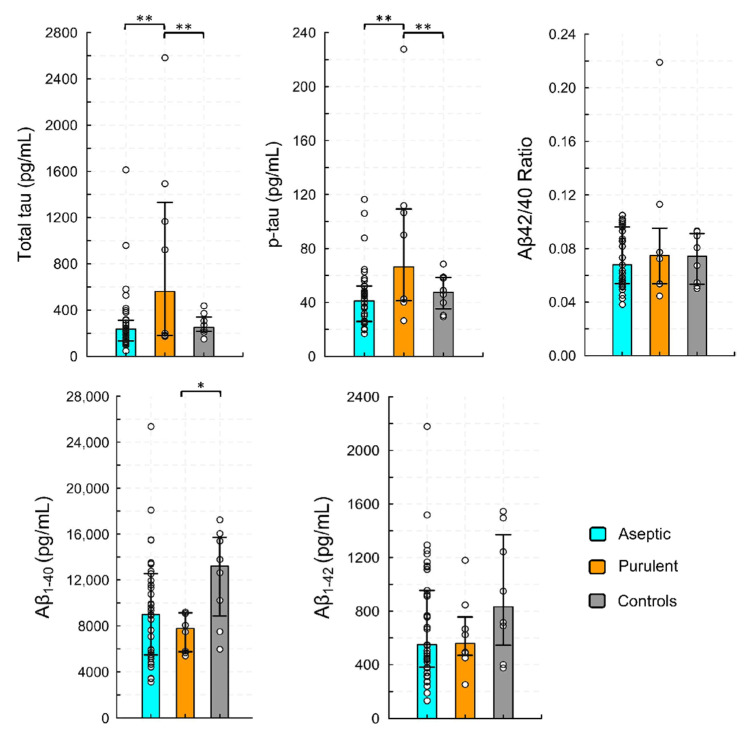
Cerebrospinal fluid concentrations of biomarkers of Alzheimer’s disease in children with infections of the central nervous system, divided by the type of infection. The aseptic group comprises children with viral infections of the CNS and children with Lyme neuroborreliosis (*n* = 37). The purulent group comprises children with purulent bacterial infections of the CNS (*n* = 8). Bars represent medians, whiskers represent interquartile range, and dots show the raw data. * indicates *p* < 0.05. ** indicates *p* < 0.01.

**Figure 3 viruses-14-00725-f003:**
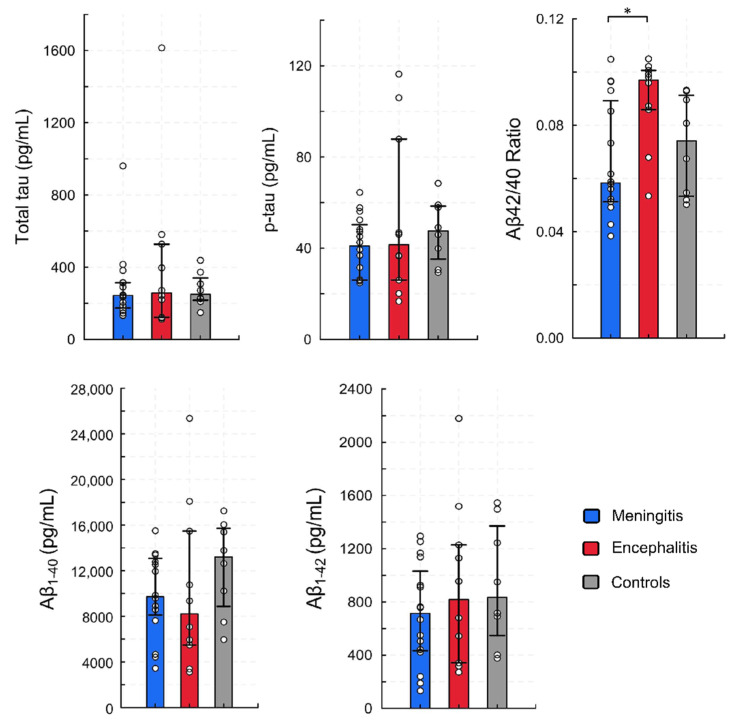
Cerebrospinal fluid concentrations of biomarkers of Alzheimer’s disease in children with infections of the central nervous system, divided by the severity of infection. In this comparison, only children with viral causes were included. The meningitis group (*n* = 16) comprises children with enteroviral meningitis (*n* = 10), varicella zoster virus meningitis (*n* = 3), and tick-borne encephalitis (*n* = 3). The encephalitis group (*n* = 10) comprises children with enteroviral encephalitis (*n* = 2), varicella zoster viral encephalitis (*n* = 2), and tick-borne encephalitis (*n* = 6). Bars represent medians, whiskers represent interquartile range, and dots show the raw data. * indicates *p* < 0.05.

**Table 1 viruses-14-00725-t001:** Clinical characteristics of the study groups.

	Enteroviruses (*n* = 12)	Varicella Zoster Virus (*n* = 5)	Tick-Borne Encephalitis Virus (*n* = 9)	Bacterial–Purulent (*n* = 8)	Bacterial–Aseptic (*n* = 11)	Controls (*n* = 8)
Female	50%	60%	44%	25%	27%	50%
Age (years)	7.3 (4.7–11.1) ^4^	13.5 (9.8–16.5) ^4^	10.6 (9.8–15.2) ^4^	2.4 (0.3–7.1) ^1,2,3,5,6^	14.9 (13.2–15.6) ^4^	8.5 (5.4–12.6) ^4^
Encephalitis	17%	40%	67%	25%	0%	-
CSF (cells/mcL)	65 (34–120) ^4^	660 (522–811) ^4^	57 (48–91) ^4^	357 (288–4011) ^1,2,3,5,6^	110 (68–250) ^4^	1 (1–2) ^4^
CSF protein (mg/dL)	26 (23.5–30) ^2,4,5^	119 (95–170) ^1,3,6^	47 (35.6–66) ^2^	59 (44–128.5) ^1,6^	80 (51–125) ^1,6^	19 (16–29) ^2,4,5^
CSF lymphocytes (%)	31 (18–40) ^2,3,5^	98 (93–98) ^1,3,4^	55 (33–88) ^1,2,4,5^	21 (17–28) ^2,3,5^	92 (86–94) ^1,3,4^	-
CSF ALC (cells/µL)	22 (10–60) ^2^	647 (485–795) ^1,3,4,5^	26 (22–63) ^2,4^	100 (63–401) ^2,3^	86 (58–243) ^2^	-
CSF neutrophils (%)	47 (26–68) ^2,3,5^	0 (0–1) ^1,4^	9 (3–35) ^1,4^	67 (45–79) ^2,3,5^	1.5 (0–5) ^1,4^	-
CSF ANC (cells/µL)	33 (24–43) ^4^	0 (0–5) ^4^	5 (1–20) ^4^	253 (157–3490) ^1,2,3,5^	2 (0–6) ^4^	-
CSF monocytes (%)	13 (3–21)	2 (2–2)	14 (6–32)	11 (3–15)	6 (3–12)	-
CSF AMC (cells/µL)	10 (1–40)	16 (7–19)	8 (3–20)	115 (39–232)	9 (2–13)	-
Serum C-reactive protein (mg/L)	11.3 (1.4–24.3) ^4^	0.2 (0.2–1) ^4^	5.7 (1.6–12.8) ^4^	204.5 (177.1–272.5) ^1,2,3,5,6^	0.6 (0.2–1.3) ^4^	4.5 (0.3–29.5) ^4^
Blood WBC (×10^3^ cells/µL)	9.7 (6.9–10.7) ^4^	6.7 (6.1–8.6) ^4^	12.4 (8.4–14.1) ^4^	22.7 (17.7–29.4) ^1,2,3,5,6^	6.8 (5.8–10.4) ^4^	7.7 (7.04–9.7) ^4^

Continuous data are presented as medians with interquartile range; categorical variables are shown as frequencies. Abbreviations: CSF, cerebrospinal fluid; ALC, absolute lymphocyte count; ANC, absolute neutrophile count; AMC, absolute monocyte count; WBC, white blood cell count; ^1^ indicates *p* < 0.05 when compared with enteroviruses; ^2^ indicates *p* < 0.05 when compared with varicella zoster virus; ^3^ indicates *p* < 0.05 when compared with tick-borne encephalitis virus; ^4^ indicates *p* < 0.05 when compared with purulent meningitis/encephalitis; ^5^ indicates *p* < 0.05 when compared with Lyme neuroborreliosis (Bacterial-Aseptic); ^6^ indicates *p* < 0.05 when compared with the control group.

**Table 2 viruses-14-00725-t002:** Cerebrospinal fluid concentrations of biomarkers of Alzheimer’s disease in study groups.

	Enteroviruses (*n* = 12)	Varicella Zoster Virus (*n* = 5)	Tick-Borne Encephalitis Virus (*n* = 9)	Bacterial–Purulent (*n* = 8)	Bacterial–Aseptic (*n* = 11)	Controls (*n* = 8)
Amyloid β_1–40_ (pg/mL)	11,362.2 (8286.4–13,087.1)	9864.8 (7081–13,506.6)	8591.3 (5477.6–9354.9)	7776.8 (5749.2–9145.9)	5905.8 (5156–11,316.1)	13,208.8 (8868.5–15,717.4)
Amyloid β_1–42_ (pg/mL)	918 (554.8–1152.7)	679.2 (506–757.3)	543.5 (317.6–955.2)	558.7 (470.1–756.6)	448.8 (375.3–755.7)	833.2 (546.6–1370.1)
Aβ42/40 ratio	0.089 (0.058–0.099)	0.056 (0.051–0.086)	0.068 (0.053–0.098)	0.075 (0.054–0.095)	0.064 (0.054–0.086)	0.074 (0.053–0.091)
Total tau (pg/mL)	243.5 (183.4–314.4) ^4^	290.6 (220.4–415.7)	243 (146.6–397.3) ^4^	562 (180.1–1331.3) ^1,3,5,6^	186.2 (99.1–243.8) ^4^	250.7 (217.7–340.3) ^4^
Phosphorylated tau (pg/mL)	46.2 (35.5–54.3)	36.6 (25.7–57.9) ^4^	36.9 (25.6–46.2)	66.4 (41.3–109.2) ^2,5^	40.4 (24.4–52) ^4^	47.5 (35.2–58.5)

Data are presented as medians with interquartile range. In this table, ^1^ indicates *p* < 0.05 when compared with enteroviruses; ^2^ indicates *p* < 0.05 when compared with varicella zoster virus; ^3^ indicates *p* < 0.05 when compared with tick-borne encephalitis virus; ^4^ indicates *p* < 0.05 when compared with purulent meningitis/encephalitis; ^5^ indicates *p* < 0.05 when compared with Lyme neuroborreliosis (Bacterial-Aseptic); ^6^ indicates *p* < 0.05 when compared with the control group.

## Data Availability

Data are within the article or Appendix A.

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
