# Peer review of "Amyloid and Tau Protein Concentrations in Children with Meningitis and Encephalitis"

_viruses, 2022, doi:10.3390/v14040725_

Round 1

Reviewer 1 Report

This is a well written and interesting paper .

However, authors should stress more the differences between groups analyzed (tables) and rewrite partially the discussion section, stressing the low number of enrolled patients, the heterogeneity and diversity of their infections, providing more possible explanations regarding each subtype of neuromarkers' alteration with regard to the involved pathogen.

Include these points in the discussion and limitations.

Commentaries:

A considerably and surprisingly high number of TBE infection was reported in this paper. Can the authors explain this finding epidemiologically or by any other mean?

Also, the same can be said for Lyme borreliosis. Can the author confirm that? Which criteria were applied? Is the regional epidemiology consistent with that finding?

The authors cite wrongly reference 22 (Sjögren, M.; Gisslén, M.; Vanmechelen, E.; Blennow, K. Low Cerebrospinal Fluid Beta-Amyloid 42 in Patients with Acute 309 Bacterial Meningitis and Normalization after Treatment. Neurosci. Lett. 2001, 314, 33–36 ). That do not refer to tuberculous meningitis but to acute bacterial meningitis instead. Maybe the authors have mistaken this paper with another (Stroffolini G, Guastamacchia G, Audagnotto S, et al. Low cerebrospinal fluid Amyloid-βeta 1-42 in patients with tuberculous meningitis. BMC Neurol. 2021 Nov 16;21(1):449. doi: 10.1186/s12883-021-02468-2.) that exactly reflects what is in the discussion section for [22], and is consistent with another paper coming from the same research lab, cited accordingly by the authors [24].

In my view, authors should also stress the particular age group (pediatrics) that was included in the analysis, and possible differences in neuromarkers compared with adults/AD patients. Also, the control group was admitted for an acute neurological syndrome (suspected meningitis): which was the final diagnosis after ruling out meningitis? 

However, the finding of altered neuromarkers in such population it is interesting: young patients may suffer from long-term neurological "signature". Did the author ruled out any other reason (HIV/pediatric neurological syndromes) for neuromarkers' alterations, or apply any other tool (EEG, neuropsicological battery testing) to further characterize neurological damage? This is a brief report, but other common neurological issues should not be overlooked (M&M section). 

Do the author have searched for complex reasons in the explanation of the findings? e.g. proteins interaction, CSF re-circulation etc... 

Author Response

We would like to thank the reviewer for all critical and constructive comments, which will help to improve the quality of the manuscript.

  1. Etiology of the study group is now explained and discussed in the manuscript.
  2. The reference is now cited correctly.
  3. To our knowledge our study is the first in pediatric age group. The discussion refers to results of adult studies what is now clearly stated in the manuscript.
  4. None of children had any underlying chronic neurological or any other condition. All children were administered to the hospital because of acute illness with sudden onset. This is stated in the Materials and Methods section now.

Reviewer 2 Report

Sulik et al. studied whether CNS infections could contribute to amyloid and tau pathology in children by measuring CSF concentrations of Aβ and tau in various viral neuroinfections and compared it with bacterial infections and controls. The major findings revealed that Aβ 42/40 concentration quotient discrepancies in CFS between meningitis and encephalitis in children, which may affect the metabolism of AD biomarkers. Overall, it is quite an interesting manuscript. There are some concerns that should take their attention

  1. Taken together, this study did not point to different mechanisms resulting in an increase of tau and phosphorylated tau concentrations levels in the CSF of patients with meningitis and encephalitis. This should be mentioned and well discussed to distinguish AD from neuroinflammatory diseases and also the metabolism of AD biomarkers.
  2. In conclusion, the authors said that infections may in fact impact amyloid metabolism, but the finding reveals no changes in CSF amyloid Aβ1– 42 concentrations. What does it impact?
  3. How those viral infectors were identified in the study while no information about it?

Author Response

We would like to thank the reviewer for all the critical and constructive comments, which will help to improve the quality of this manuscript.

Taken together, this study did not point to different mechanisms resulting in an increase of tau and phosphorylated tau concentrations levels in the CSF of patients with meningitis and encephalitis. This should be mentioned and well discussed to distinguish AD from neuroinflammatory diseases and also the metabolism of AD biomarkers.

We have modified the discussion section in order to point to different mechanisms affecting tau and amyloid levels.

In conclusion, the authors said that infections may in fact impact amyloid metabolism, but the finding reveals no changes in CSF amyloid Aβ1– 42 concentrations. What does it impact?

We found decreased Aβ1– 40 concentrations in purulent infections, compared to controls, and decreased Aβ42/40 ratio in meningitis, compared to encephalitis. The text was modified for clarity.

How those viral infectors were identified in the study while no information about it?

The information about diagnostic tests is now included in the Materials and Methods section.